# Serum Leucine-Rich α2 Glycoprotein: A Biomarker for Predicting the Presence of Ulcerative Colitis but Not Ulcerative Proctitis

**DOI:** 10.3390/jcm11216366

**Published:** 2022-10-28

**Authors:** Ichitaro Horiuchi, Akira Horiuchi, Takeji Umemura

**Affiliations:** 1Department of Gastroenterology, Shinshu University Hospital, Matsumoto 390-8621, Japan; 2Digestive Disease Center, Showa Inan General Hospital, Komagane 399-4117, Japan

**Keywords:** leucine-rich α2 glycoprotein, biomarker, ulcerative colitis

## Abstract

The serum level of leucine-rich α2 glycoprotein (LRG) is a biomarker for active ulcerative colitis (UC). We examined the serum level of LRG as a biomarker for predicting the presence of UC. Patients with persistent diarrhea and/or bloody stool with no history of UC were consecutively enrolled at their initial visit. Serum LRG measurement and colonoscopy with histology were performed on the same day. We enrolled 103 patients (69 men; median age, 45 years) with suspected UC; 66 patients were diagnosed with active UC (proctitis, n = 10; left-sided colitis, n = 26; and pancolitis, n = 30) based on endoscopic and histological criteria. Although the median LRG value in patients with proctitis was similar to that of patients with normal colonoscopic findings (8.5 vs. 8.6 mg/mL, *p* = 0.24), the median LRG values were significantly elevated in patients with left-sided colitis and pancolitis compared with those of patients with normal colonoscopy (13.6 or 18.0 vs. 8.6 mg/mL, *p* < 0.0001). The LRG cut-off value of 10.8 μg/mL was derived from the ROC curve, showing 96% sensitivity and 97% specificity for active UC but not active proctitis. Using a cut-off value of 10.8 mg/mL serum, LRG could be a novel biomarker for predicting patients with active UC except for proctitis.

## 1. Introduction

Inflammatory bowel diseases (IBD), comprising Crohn’s disease and ulcerative colitis (UC), are characterized by chronic inflammation of the intestinal mucosa of unknown etiology, and the number of patients with IBD is increasing worldwide [1]. UC has recently become more common in Japan [2]. Although UC is typically diagnosed based on colonoscopic and histological findings, a colonoscopy that remains highly uncomfortable for patients requires bowel preparation and typically must be scheduled. In addition, expert gastroenterologists and pathologists are required for a definitive diagnosis of UC, which relies on the histological assessment of bowel biopsies from colonoscopy. A diagnostic delay may cause a delay in appropriate therapies, resulting in disease progression and an increased risk of complications. Therefore, an easily accessible serum biomarker could be an important tool for predicting the presence of active UC.

Serada et al. identified leucine-rich α2 glycoprotein (LRG) as a novel biomarker for rheumatoid arthritis and IBD using a proteomic approach [3]. An LRG is a 50 kDa glycoprotein that contains repetitive sequences with a leucine-rich motif [4]. An LRG was induced by interleukin-22, tumor necrosis factor-α, and interleukin-1β, independently from interleukin-6 [3]. It was reported in Japan that serum levels of LRG are closely correlated with endoscopic activity in UC, with a cut-off value of 16 μg/mL for the presence of active UC [5,6]. However, the role of LRG testing in ulcerative proctitis is unclear compared to left-sided colitis or pancolitis [6,7]. We conducted the present prospective study to investigate the usefulness of LRG as a serum biomarker for predicting the presence of active UC and/or ulcerative proctitis.

## 2. Materials and Methods

### 2.1. Study Design

This was a prospective, observational cohort study performed at the Showa Inan General Hospital in Komagane, Japan. The study was conducted in accordance with the latest version of the Helsinki Declaration and approved by the Showa Inan General Hospital’s Ethics Committee on 23 September 2019 (2019-08). In addition, we confirmed that all examinations in this study were performed in accordance with relevant guidelines/regulations [8]. All subjects gave written informed consent when the enrollment for this study was scheduled. 

### 2.2. Patients

Between October 2020 and December 2021, 1121 patients visited an outpatient clinic at our department of Showa Inan General Hospital. Of these 125 patients, had persistent diarrhea and/or bloody stool without a prior history of UC and were enrolled in this study as patients with suspected UC. The exclusion criteria were the presence of acute or chronic renal failure, chronic heart disease, pulmonary infection, liver cirrhosis, colorectal cancer, autoimmune disease, ischemic colitis, and infectious colitis. After a colonoscopy with histology, patients with possible ischemic colitis, Crohn’s disease, or intermediate colitis were excluded.

Figure 1 is the flowchart of the patient enrollment. A total of 125 patients were consecutively included on their first visit to our outpatient clinic. Fifteen patients were excluded (ischemic colitis, n = 8 patients; infectious colitis, n = 5 patients; and colorectal cancer, n = 2 patients). After colonoscopy, seven patients with possible ischemic colitis were excluded. A final total of 103 patients (67% male; median age of 45 years) were enrolled in this study.

### 2.3. Measurement of Serum C-Reactive Protein (CRP) and LRG Levels

Blood sampling and colonoscopy with histology were performed on the same day, shortly after the patients’ first visit to our outpatient clinic. CRP was routinely measured. Serum LRG was measured using a commercially available kit (Nanopia LRG, Sekisui Medical, Tokyo, Japan) at SRL, Inc. (Tokyo, Japan) by a latex agglutination method using antihuman LRG mouse monoclonal antibody, as described in [5,6].

### 2.4. Assessment of UC

For the evaluation of UC, clinical activity was graded using the partial Mayo score: active disease was defined as a score of ≥2. The endoscopic activity was graded using the Mayo endoscopic sub-score in patients with active UC (score ≥ 1) [9]. Patients with UC were classified as having proctitis, left-sided colitis, or pancolitis according to the extent of their disease, as described by the Montreal classification [10]. Histologic studies were evaluated using the Geboes score [11] by a certificated pathologist. According to the scoring system, the histologic disease activity of UC was classified into 6 grades, with grades from 0 to 5. For cases with endoscopic activity, the biopsy specimen from the site with maximum endoscopic activity was evaluated. 

### 2.5. Statistical Analysis

When we estimated the minimum expected area under the curve to be 0.7 and the ratio between cases without UC and those with UC to be 1:2, the required sample size was calculated to be at least 80 with 80% power. Data are presented as means and standard deviations. The χ^2^-test (with Yates’s correction for continuity where appropriate) was used for the comparison of categorical data. Fisher’s exact test was used when the numbers were small. For parametric data, a Student’s *t*-test was used when two means were compared. Non-parametric data were analyzed by the Mann–Whitney U-test when two medians were compared. The relationship between the presence of active UC and partial Mayo score, CRP, or LRG was examined using receiver operating characteristic (ROC) curves and the area under the curve (AUC). A probability *p*-value < 0.05 was considered significant. The statistical analyses were performed using GraphPad Prism ver. 9.3.1 (GraphPad Software, La Jolla, CA, USA).

## 3. Results

### 3.1. Characteristics of the Enrolled Patients

He patients’ characteristics and their clinical findings are summarized in Table 1. The median age of the patients was 45 (range 18–79) years. Although only a 79-year-old man was enrolled, a higher level of active UC was not observed in the elderly patients in this study. The median duration of symptoms prior to diagnosis of UC was 7 days (range 3–40 days). No medications were used for their symptoms prior to undergoing colonoscopy, because this procedure was performed shortly after the patients’ visits to our outpatient clinic. The median partial Mayo score was 4 (range 1–7). The median CRP was 0.1 (range 0.01–4.99) mg/dL. The median LRG was 11.1 (range 5.8–50.8) μg/mL.

Table 2 shows enrolled subjects’ endoscopic and histological findings. Normal colonoscopic findings were observed in 37 patients (36%); 66 patients had active UC: proctitis, n = 10 patients (10%); left-sided colitis, n = 26 patients (25%); and pancolitis, n = 30 patients (29%). Mayo endoscopic subscores of 0, 1, 2, and 3 were 37 (36%), 26 (25%), 37 (36%), and 3 (3%), respectively. In addition, Geboes scores of 0, 1, 2, 3, 4, 5, and 6, were 37 (36%), 21 (20%), 27 (26%), 8 (8%), 7 (7%), and 3 (3%), respectively.

### 3.2. Clinical Outcomes of Enrolled Patients

Although the median LRG value in the 10 patients with proctitis was similar to that of the 37 patients with normal colonoscopic findings (8.5 vs. 8.6 mg/mL, *p* = 0.24), the median LRG values were significantly elevated in the 56 patients with left-sided colitis or pancolitis compared with those of normal colonoscopy (13.6 or 18.0 vs. 8.6 mg/mL, *p* < 0.0001) (Figure 2).

Table 3 shows the results of the comparison of each value of partial Mayo score, CRP, and LRG in the patients with normal colonoscopic findings (controls) and patients with active UC (with the exception of those with ulcerative proctitis). The median partial Mayo score, CRP, and LRG values were obtained. The patients with active UC have significantly higher levels compared to the patients with normal colonoscopic findings: partial Mayo score 5 vs. 3, *p* < 0.0001; CRP 0.26 vs. 0.02, *p* < 0.0001; and LRG 15.9 vs. 8.5, *p* < 0.0001.

### 3.3. ROC Curves for Partial Mayo Score, CRP, and LRG in Predicting Active UC Excluding Patients with Only Proctitis

We evaluated the diagnostic ability of LRG for detecting the presence of active UC compared with the partial Mayo score and CRP by analyzing the area under the curve (AUC) of the ROC curves (Figure 3). The AUC for LRG was significantly higher than those of both the partial Mayo score (*p* < 0.0001) and CRP (*p* < 0.0001). Using a cut-off value of 10.8 μg/mL, which was derived from the ROC curve, LRG showed markedly higher sensitivity (96%) and specificity (97%) than CRP and the partial Mayo score (Table 4). These findings show that LRG levels, even in patients with normal CRP levels, were more sensitive than CRP levels in predicting the presence of active UC (with the exception of ulcerative proctitis).

## 4. Discussion

We evaluated the usefulness of the serum LRG for identifying the presence of active UC (with the exception of ulcerative proctitis) in patients presenting with a clinical picture suggestive of active colonic inflammatory bowel disease. Previous reports indicated that the serum LRG was a potential biomarker for monitoring disease activity in UC [12,13,14]. The LRG was also described as a highly sensitive serum biomarker for detecting small bowel mucosal activity in patients with Crohn’s disease, at a cut-off value of 13.4 μg/mL [15]. In the present study, the serum LRG was elevated to >10.8 μg/mL in patients with left-sided colitis or the pancolitis type of active UC compared to patients with normal colonoscopic findings. Therefore, our findings confirm that the serum LRG with a cut-off value of 10.8 μg/mL is a good biomarker for predicting the presence of active UC (left-sided colitis or pancolitis) in patients presenting with symptoms suggestive of active colonic inflammation. LRG was also superior to the partial Mayo score and CRP.

This study also revealed the limitation of serum LRG for predicting all types of active UC, as the serum LRG was not elevated (i.e., was <10 mg/mL) in the 10 patients with ulcerative proctitis. As reflected in Figure 2, we suspect that the degree of the elevation of serum LRG might be closely related to the extent of the colorectal mucosa with active inflammation, with the result that LRG was not elevated in our patients with proctitis compared to patients with left-sided colitis or pancolitis (median LRG 8.5 vs. 13.6 and 18.0 mg/mL, *p* < 0.0001). In addition, the median LRG value in the patients with pancolitis was significantly higher than that of patients with left-sided colitis (*p* = 0.0003). Therefore, even if the serum LRG is <10 μg/mL in patients with persistent diarrhea and/or bloody stool with no history of UC, the proctitis type of UC should not be excluded. We may be able to predict the extent of the colorectal mucosa with active UC using the LRG value. 

In the present study, a predictive diagnostic model based on a ROC curve assessed the capability of LRG to discriminate against active UC except for ulcerative proctitis and healthy groups. A cut-off value of 10.8 μg/mL was derived from the ROC curve (Figure 3C). In our tested population, LRG was able to identify real UC cases with a sensitivity of 96% and a specificity of 97% (Table 4). These results suggest that LRG could be a valuable, non-invasive serum biomarker to triage patients suspected of having UC but not ulcerative proctitis in a primary care facility. The future clinical potential of LRG may be intended to accelerate early diagnosis in patients with active UC except for ulcerative proctitis.

It is difficult to diagnose active UC without a colonoscopy and histology. In the present evaluation, CRP showed low sensitivity (48%). The results confirm that the CRP level could not be utilized to predict the presence of active UC. On the other hand, calprotectin is a calcium- and zinc-binding protein mainly produced by neutrophils. It has been reported that fecal calprotectin levels reflect the local inflammation of the gastrointestinal tract. The fecal calprotectin level has been reported as a reliable surrogate marker of endoscopic and histologic activity in UC [16,17]. Recently, researchers reported the association between fecal calprotectin levels and laboratory data (CRP, ESR, serum albumin, hemoglobin, and platelets), as well as endoscopic and histologic disease activity in Japanese UC patients [18]. The results show that fecal calprotectin is closely related to endoscopic and histologic disease activity and clinical data. In this study, we did not measure fecal calprotectin because fecal samples are associated with inherent problems such as limited quantitative capability and cumbersome sampling. Although we could not perform a direct comparison with fecal calprotectin in the present study, our results demonstrated that the use of the serum LRG was both convenient and practical, requiring only a blood sample.

In a study of 81 IBD patients, including 47 UC patients, the serum LRG levels decreased along with improvements in clinical and endoscopic outcomes upon adalimumab treatment (27.4 ± 12.6 mg/mL at week 0, 15.5 ± 7.7 mg/mL at week 12, 15.7 ± 9.6 mg/mL at week 24, and 14.5 ± 6.8 mg/mL at week 52), being correlated with endoscopic activity at each time point [12]. The LRG levels in the remission of UC patients might be less than 16 mg/mL.

This study has some limitations. It was conducted at a single hospital in Japan, and the number of enrolled patients was small (n = 103). It was also not possible to eliminate the influence of the patients’ backgrounds or patient selection bias. Larger, randomized multicenter trials including many patients are necessary to test our findings.

## 5. Conclusions

Measuring serum LRG using a cut-off value of 10.8 mg/mL could be a useful novel biomarker for predicting patients with active UC other than proctitis.

## Figures and Tables

**Figure 1 jcm-11-06366-f001:**
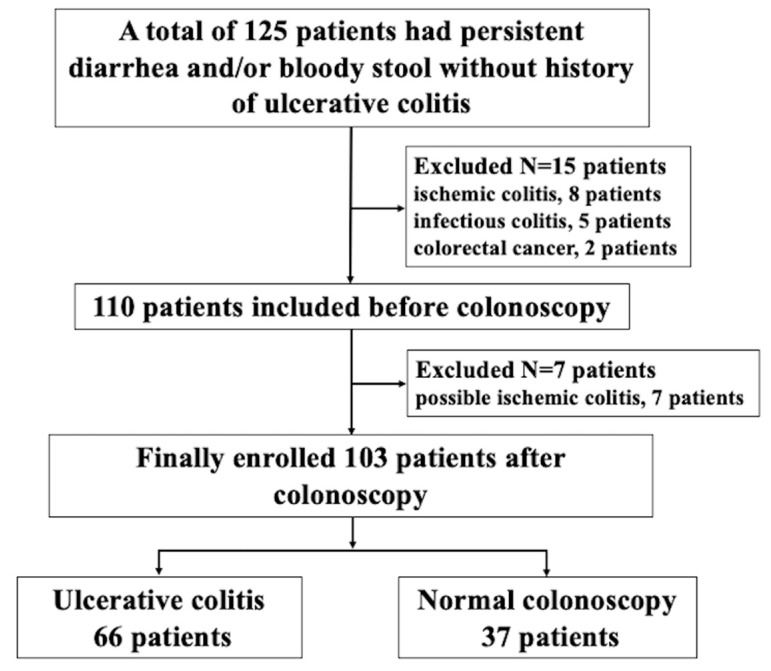
The flowchart of enrolled patients.

**Figure 2 jcm-11-06366-f002:**
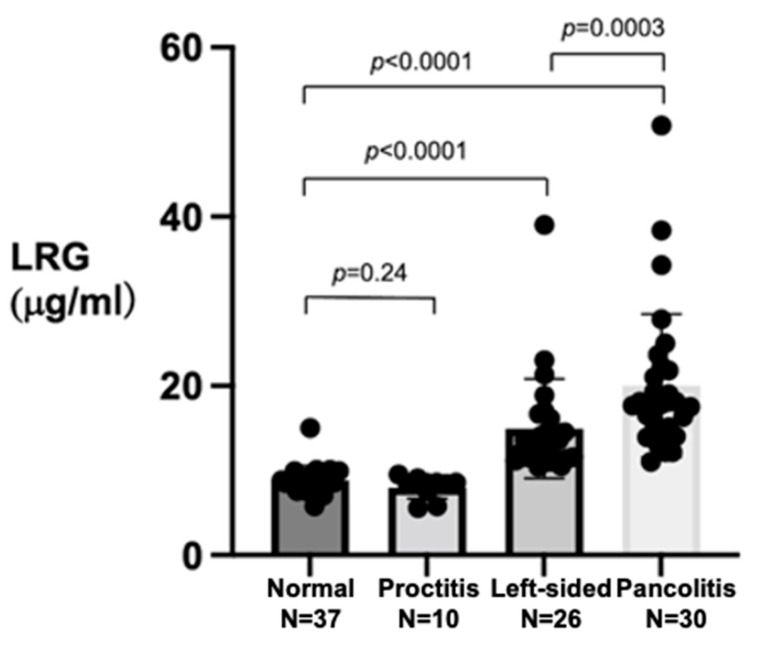
Comparison of median leucine-rich α2 glycoprotein (LRG) values among patients with normal colonoscopic findings, ulcerative proctitis, left-sided colitis, and pancolitis.

**Figure 3 jcm-11-06366-f003:**
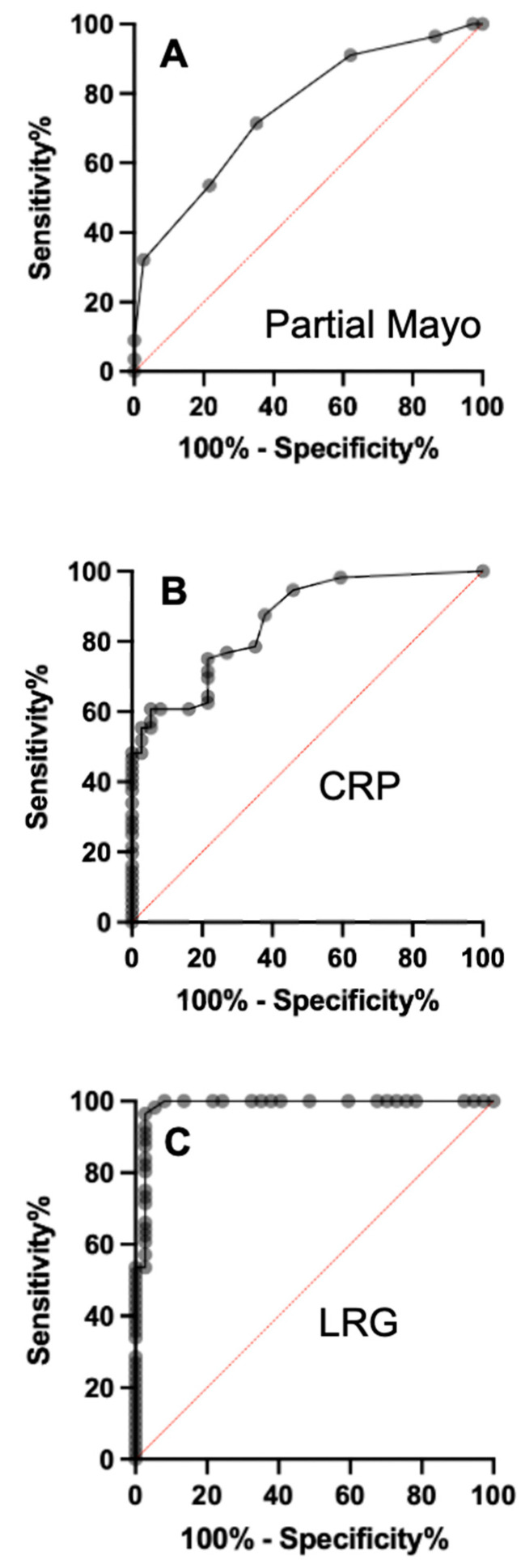
ROC curves for partial Mayo scores (**A**), CRP (**B**), and LRG (**C**) for predicting active ulcerative colitis other than proctitis.

**Table 1 jcm-11-06366-t001:** Enrolled subjects’ characteristics and their clinical findings (n = 103).

Males/females, n (%)	69 (67)/34 (23)
Age, years (range)	45 (18–79)
Median duration of symptoms, days (range)	7 (3–40)
Median partial Mayo score, (range)	4 (1–7)
Median white blood cell count, 10^9^/L (range)	6.7 (4.4–10.7)
Median hemoglobin, g/dL (range)	12.2 (10.3–13.5)
Median platelet count, 10^9^/dL (range)	32.5 (22–37.5)
Median albumin, g/dL (range)	3.4 (2.9–4.2)
Median CRP, mg/dL (range)	0.1 (0.01–4.99)
Median ESR, mm/h (range)	24 (11–46.5)
Median LRG, mg/mL(range)	11.1 (5.8–50.8)

CRP: C-reactive protein, ESR: erythrocyte sedimentation rate, LRG: leucine-rich α2 glycoprotein.

**Table 2 jcm-11-06366-t002:** Enrolled subjects’ endoscopic findings and histological findings (n = 103).

Location of Colonoscopy Findings, n (%)	
Normal	37 (36)
Proctitis	10 (10)
Left-sided colitis	26 (25)
Pancolitis	30 (29)
Endoscopic findings of the Mayo endoscopic subscore, n (%)	
0	37 (36)
1	26 (25)
2	37 (36)
3	3 (3)
Histological findings of the Geboes score, n (%)	
0	37 (36)
1	21 (20)
2	27 (26)
3	8 (8)
4	7 (7)
5	3 (3)

**Table 3 jcm-11-06366-t003:** Comparison of each value of partial Mayo score, C-reactive protein (CRP), and leucine-rich α2 glycoprotein (LRG) between the control patients with normal colonoscopic findings and those with ulcerative colitis other than proctitis.

	Control (n = 37)	UC (n = 56)	*p* Value
Median partial Mayo score (range)	3 (1–6)	5 (1–7)	<0.0001
Median CRP, mg/dL (range)	0.02 (0.01–0.2)	0.26 (0.15–4.99)	<0.0001
Median LRG, mg/mL (range)	8.6 (5.8–15)	15.9 (11–50.8)	<0.0001

UC: ulcerative colitis other than proctitis, CRP: C-reactive protein, LRG: leucine-rich α2 glycoprotein.

**Table 4 jcm-11-06366-t004:** Area under the curve (AUC), sensitivity, and specificity of partial Mayo score, CRP, and LRG for predicting ulcerative colitis except proctitis.

Variable	AUC (95% CI)	Cut-Off Value	Sensitivity,% (95% CI)	Specificity,% (95% CI)
Partial Mayo	0.75 (0.65–0.85)	3.5	71 (58.5–81.6)	65 (48.8–78.1)
CRP	0.86 (0.79–0.93)	0.27 mg/dL	48 (35.7–61)	97 (86.2–99.9)
LRG	0.99 (0.96–1.0)	10.8 μg/mL	96 (87.9–99.3)	97 (86.2–99.9)

AUC: Area under the curve, CRP: C-reactive protein, LRG: leucine-rich α2 glycoprotein.

## Data Availability

The data underlying this article will be shared upon reasonable request to the corresponding author.

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
