# Peer review of "Serum Leucine-Rich α2 Glycoprotein: A Biomarker for Predicting the Presence of Ulcerative Colitis but Not Ulcerative Proctitis"

_jcm, 2022, doi:10.3390/jcm11216366_

Round 1

Reviewer 1 Report

Manuscript ID: jcm-1950296

Type of manuscript: Article Title: Serum leucine-rich a2 glycoprotein: A biomarker for predicting the presence of ulcerative colitis but not ulcerative proctitis

Authors: Ichitaro Horiuchi, Akira Horiuchi *, Takeji Umemura

Ulcerative colitis (UC) is a common disorder characterized by recurring episodes of inflammation limited to the mucosal layer of the colon. Generally, it involves the rectum and may extend in a proximal and continuous fashion to involve other parts of the colon. Although UC primarily involves the bowel, it is associated with manifestations in other organs and is related to extraintestinal manifestations, that tend to follow the clinical course of the colitis. Horiuchi et al., investigated the effectiveness of the serum level of leucine-rich alpha 2 glycoprotein (LRG) has a biomarker for active UC. The authors by examining the serum LRG found a new cutoff value, estimated around 10.8 ug/mL, for predicting patients with active UC except for proctitis.

The manuscript is interesting but is missing some information and sometimes the organization is not logical. Comments that are required to be addressed are found below.

1. The Introduction must be improved including a little bit of background on inflammatory bowel diseases, the frequency of UC also in other Countries, since it is not limited to Japan. More description is also required for LRG and the other inflammatory processes the authors considered in the manuscript, as proctitis.

2. References are lacking, as for example pag.1, line 42.

3. Pag.2, lines 44-45: I would move this text at the end, in the “Institutional Review Board Statement.

4. In the Material & Methods the authors at pag. 2 line 55 introduce blood sampling and colonoscopy: it would be great to add the data, especially the one refers to colonoscopy or any other clinical and/or histological assessment. Colonoscopy and/or Histology assessment is highly required in the manuscript.

5. I would suggest to present Figures and Tables as they appear in the manuscript. For example, Table 1 and Table 2 appear before Figure 1, that is introduced at pag. 2 line 80. 

6. When possible, I would suggest to provide information for males and females separately.

7. The age range is very broad: have the authors observed active UC more in elders?

8. In Figure 2 it would be possible to include “n” the number of subjects in the X axis?

9. I would suggest to move the p value in the legend since it is the same for all the three graphs. Perhaps, the information of the three graphs can be summarize in a table. What should I read in Y axis? Please, specify it.

10. In the legend of Figure 4, I would change “upper”, “middle” and “lower” with “A”, “B”, and “C”, that then would need to be added to the corresponding panel in Figure 4.

11. The Discussion should be improved. In addition, the authors should emphasize the finding of a new cut-off for predicting patients with active UC.

12. At pag. 6 line 165, the authors speak about inflammation. It would be great if other inflammatory markers have been evaluated. If so, the information should be included in the main manuscript.

13. Do the authors have information about the remission of the patients and how LRG levels may change with the remission time? Please, include and/or comment in the Discussion.

Reviewer 2 Report

1. The introduction does a poor job of explaining the need for a non-invasive biomarker for diagnosing UC. More detail regarding the unmet clinical need for diagnostic biomarkers for IBD should be presented.

2. Study Design, Inclusion, and Exclusion Criteria:

what measures were taken to exclude Crohn's disease or indeterminate colitis patients?

What cut-off was utilized for CRP levels?

Since this is a prospective study, please show proof of your calculations that the current sample size confers enough power analysis to show that this study is not underpowered.

3. Median disease duration is reported, however, a diagnosis of UC was not made prior, I think it would be better to report this variable as the duration of symptoms prior to diagnosis of UC.

4.  Can you please provide details of medications used by patients for their symptoms prior to undergoing colonoscopy?

5. What histologic index was used to diagnose UC?

6. Considering the non-specific nature of CRP, comparing LRG to fecal calprotectin would be a much stronger indicator of UC.

Round 2

Reviewer 1 Report

The manuscript has improved and the authors have addressed all the major concerns.

Reviewer 2 Report

The authors have answered all my concerns and made significant changes accordingly.